# CELA3B immunostaining is a highly specific marker for acinar cell carcinoma of the pancreas

Ria Uhlig[1], Nina Bröker[1], Sören Weidemann[1], Natalia Gorbokon[1], Anne Menz[1], Franziska Büscheck[1], Andreas M. Luebke[1], Devita Putri[1], Martina Kluth[1], Claudia Hube-Magg[1], Andrea Hinsch[1], Maximilian Lennartz[1], Viktor Reiswich[1], Doris Höflmayer[1], Christoph Fraune[1], Katharina Möller[1], Christian Bernreuther[1], Patrick Lebok[1], Guido Sauter[1], Sarah Minner[1], Stefan Steurer[1], Eike Burandt[1], Rainer Krech[2], David Dum[1], Andreas Marx[1,3], Ronald Simon[1]*, Till Krech[1,2], Till S. Clauditz[1]☯, Frank Jacobsen[1]☯

1 Institute of Pathology, University Medical Center Hamburg-Eppendorf, Hamburg, Germany, 2 Institute of Pathology, Clinical Center Osnabrueck, Osnabrueck, Germany, 3 Department of Pathology, Academic Hospital Fuerth, Fuerth, Germany

☯ These authors contributed equally to this work.

* r.simon@uke.de

**Data Availability Statement:** All data relevant to the study are included in the article.

## Abstract

Chymotrypsin-like elastase family member 3B (CELA3B, elastase-3B) is a pancreatic enzyme with digestive function in the intestine. Since RNA analyses of normal tissues suggest that CELA3B expression is limited to the pancreas, the potential diagnostic utility of CELA3B immunohistochemistry for the distinction of pancreatic from extrapancreatic cancers and in the distinction of acinar cell carcinoma from ductal adenocarcinoma was assessed. CELA3B expression was successfully analyzed in 13,223 tumor samples from 132 different tumor types and subtypes as well as 8 samples each of 76 different normal tissue types by immunohistochemistry in a tissue microarray format (TMA). In normal tissues, CELA3B immunostaining was only seen in acinar cells and in a fraction of ductal cells of the pancreas as well as on some apical membranes of surface epithelial cells of the intestine. Among tumors, CELA3B immunostaining was seen in 12 of 16 (75%) acinar cell carcinoma of the pancreas including 6 cases with strong staining (37.5%) as well as in 5 of 13,207 other tumors (0.04%). These included 1.2% of 91 adenoid cystic carcinomas, 1.2% of 246 mucoepidermoid carcinomas and 0.8% of 127 acinic cell carcinomas of salivary glands. Our data show a good sensitivity (75%) and a high specificity (99.9%) of CELA3B immunohistochemistry for diagnosing acinar cell carcinoma of the pancreas.

## Introduction

Chymotrypsin-like elastase family member 3B also known as elastase-3B, is a 29 kDa enzyme that is encoded by the CELA3B gene located at 1p36.12. Elastases form a six member subfamily of serine proteases that hydrolyze elastin and other proteins. In contrast to its name, Elastase 3B has little elastolytic activity [1–4]. Elastase 3B is secreted from the pancreas as a zymogen

**Funding:** The author(s) received no specific funding for this work.

**Competing interests:** The CELA3B antibody clone MSVA-410M was provided from MS Validated Antibodies GmbH (owned by a family member of GS). This does not alter our adherence to PLOS ONE policies on sharing data and materials.

and has a digestive function in the intestine [5, 6]. Elastase 3B preferentially cleaves proteins after alanine residues [4]. It also plays a role in the intestinal transport and metabolism of cholesterol [6–8]. Quantification of the fecal excretion of elastase 3B is commonly used to measure the pancreatic function in clinical assays [1, 9–11].

A role of CELA3B in cancer has recently been proposed due to results from genome- and transcriptome-wide association studies as well as large-scale next generation sequencing studies. Zhong et al. compared data from 9,040 pancreatic cancer cases and 12,496 controls and found that downregulation of CELA3B was statistically linked to a higher risk of cancer [12]. Moore et al found a rare mutation of CELA3B which results in upregulation of the gene to be linked to hereditary pancreatitis and pancreatic adenocarcinoma [13]. Zhou et al utilized a RNA expression dataset derived from "The Cancer Genome Atlas" (TCGA; https://www.cancer.gov/tcga) to study the prognostic impact of gene expression on the patient outcome in 364 patients with serous cystadenocarcinoma of the ovary and identified CELA3B as one of three genes that jointly enabled a stratification in low and high risk groups [14]. Moreover, pan-cancer RNA expression data available from the cBioPortal for cancer genomics [15–17] suggest that CELA3B can be occasionally expressed also in other tumor types.

Since RNA based analyses of normal tissues suggest that CELA3B expression is completely limited to the pancreas, CELA3B antibodies may assist in the distinction of pancreatic from extrapancreatic cancers and perhaps also in the distinction of acinar cell carcinoma from ductal adenocarcinoma of the pancreas. However, immunohistochemical analyses of CELA3B expression in tumors are so far lacking. To assess the diagnostic utility of CELA3B expression analysis, the protein was evaluated in more than 15,000 tumor tissue samples from 132 different tumor types and subtypes as well as 76 non-neoplastic tissue categories by immunohistochemistry (IHC) in a tissue microarray format in this study.

## Methods

### Tissue Microarrays (TMAs)

The normal tissue TMA used in this study contained 8 samples from 8 different donors for each of 76 different normal tissue types (608 samples on one slide). The cancer TMAs included 15,099 primary tumors from 132 tumor types and subtypes. The composition of both normal and cancer TMAs is described in detail in the results section. All samples were from the archives of the Institutes of Pathology, University Hospital of Hamburg, Germany, the Institute of Pathology, Clinical Center Osnabrueck, Germany, and Department of Pathology, Academic Hospital Fuerth, Germany. Tissues were fixed in 4% buffered formalin and then embedded in paraffin. TMA tissue spot diameter was 0.6 mm. The use of archived remnants of diagnostic tissues for manufacturing of TMAs and their analysis for research purposes as well as patient data analysis has been approved by local laws (HmbKHG, §12) and by the local ethics committee (Ethics commission Hamburg, WF-049/09). All work has been carried out in compliance with the Helsinki Declaration. Data on CPA1 expression were available for 10,334 of our tumors from a previous study [18].

### Immunohistochemistry

Freshly cut TMA sections were immunostained on one day and in one experiment. Slides were deparaffinized and exposed to heat-induced antigen retrieval for 5 minutes in an autoclave at 121˚C in pH 7,8 buffer. Primary antibodies specific for CELA3B (rabbit recombinant, MS Validated Antibodies, MSVA-410M) were applied at 37˚C for 60 minutes at a dilution of 1:1800. Bound antibody was then visualized using the EnVision Kit™ (Agilent, CA, USA; #K5007) according to the manufacturer's directions. For tumor tissues, the percentage of

positive neoplastic cells was estimated, and the staining intensity was semiquantitatively recorded (0, 1+, 2+, 3+). For statistical analyses, the staining results were categorized into four groups for each antibody. Tumors without any staining were considered negative. Tumors with 1+ staining intensity in ≤70% of tumor cells or 2+ intensity in ≤30% of tumor cells were considered weakly positive. Tumors with 1+ staining intensity in >70% of tumor cells, 2 + intensity in 31–70%, or 3+ intensity in ≤30% were considered moderately positive. Tumors with 2+ intensity in >70% or 3+ intensity in >30% of cells werde considered strongly positive.

## Results

### CELA3B in normal tissues

In the pancreas, a strong cytoplasmic CELA3B immunostaining was seen in all acinar cells and in a fraction of ductal cells. In the small intestine and the colorectum, a distinct staining of the apical membranes of surface epithelial cells was seen in a fraction of samples at variable intensity (Fig 1). CELA3B staining was completely absent in striated muscle, heart muscle, smooth muscle, myometrium of the uterus, corpus spongiosum of the penis, ovarian stroma, fat, skin, hair follicle, oral mucosa of the lip, oral cavity, surface epithelium of the tonsil, transitional mucosa of the anal canal, ectocervix, squamous epithelium of the esophagus, urothelium of the renal pelvis and urinary bladder, decidua, placental trophoblastic cells, lymph node, spleen, thymus, tonsil, surface mucosa of the stomach and the gall bladder, liver, parotid gland, submandibular gland, sublingual gland, kidney, prostate, seminal vesicle, epididymis, testis, respiratory epithelium, lung, breast, endocervix, fallopian tube, corpus luteum and follicular cyst of the ovary, parathyroid gland, cerebellum, cerebrum and the pituitary gland.

### CELA3B in cancer

A CELA3B immunostaining was observed in 17 of 13,223 successfully analyzed tumors which all belonged to only 4 of 132 analyzed tumor categories (Table 1). A cytoplasmatic CELA3B positivity was most commonly seen in acinar cell carcinoma of the pancreas (75% of 16 spots positive, 37% strongly positive) (Fig 2). A weak to moderate cytoplasmatic granular staining was observed in individual cases of salivary gland tumors (3 mucoepidermoid carcinoma, 1 adenoid cystic carcinoma and 1 acinic cell carcinoma). Images of all 5 "non-pancreatic" cases with positive CELA3B immunostaining are shown in Fig 3. CELA3B immunostaining was completely absent in a further 13,202 evaluable tumors from 128 different cancer types and subtypes. For detection of acinar cell carcinoma, these figures (12 correct positive cases, 5 false positive cases, 4 false negative cases, and 13,202 correct negative cases) result in a sensitivity of 75% and a specificity of 99.9%. The extent of CELA3B immunostaining was significantly linked to the level of CPA1 expression in 16 pancreatic acinar cell cancers (p<0.0001; Fig 4) for which data were available from an earlier study [19].

## Discussion

Considering the large scale of our study, emphasis was placed on the appropriate validation of our CELA3B immunohistochemistry assay. Based on recommendations of the International Working Group for Antibody Validation (IWGAV) we compared our CELA3B staining data with expression data obtained by another independent method [23]. Normal tissue RNA expression data derived from three different publicly accessible databases [22, 24–26] were therefore compared with immunostaining results in 76 different normal tissues categories. This broad range of tissues is likely to contain most proteins that are normally expressed at relevant levels in cells of adult humans and should therefore enable the detection of most

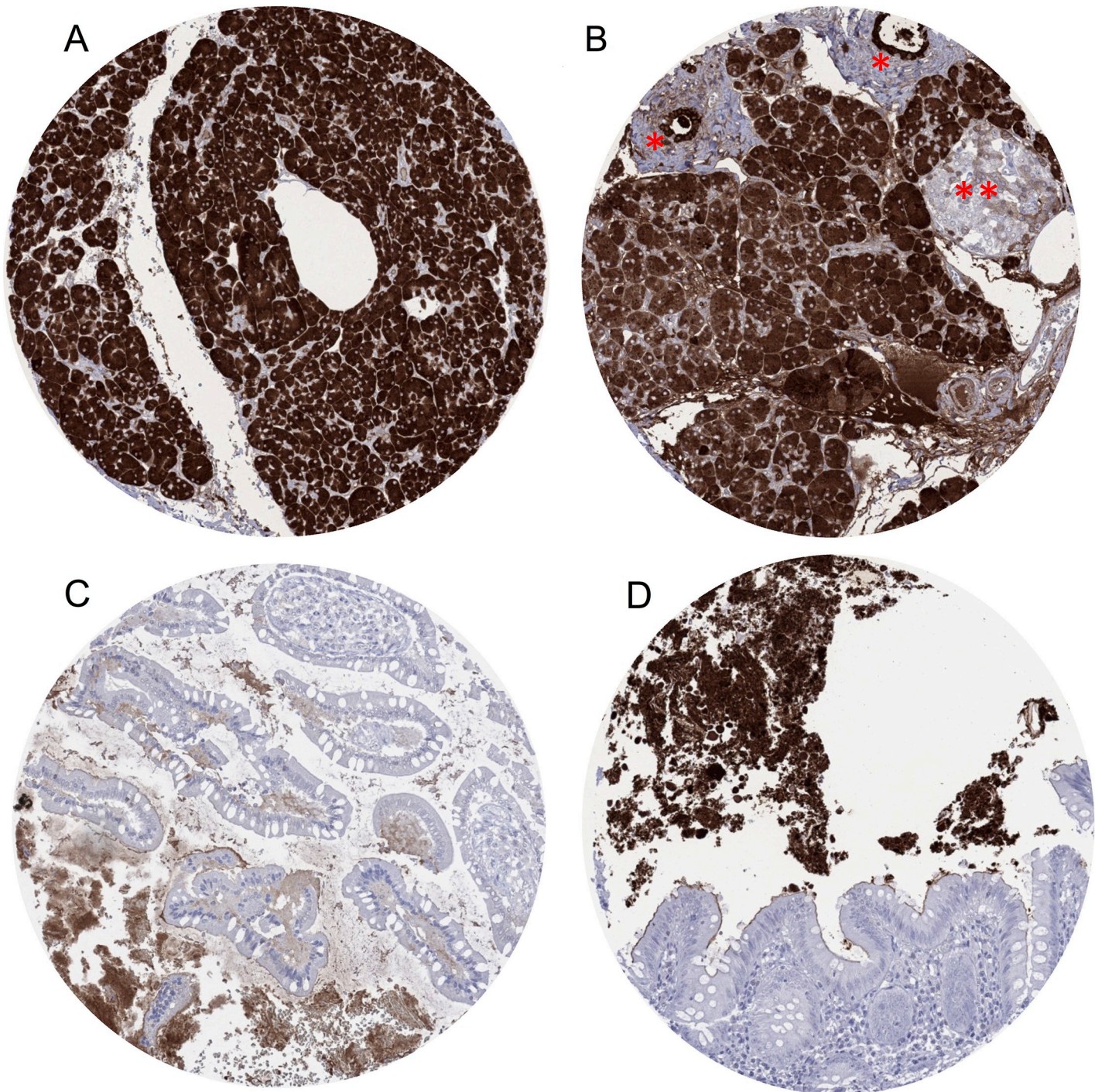

**Fig 1. CELA3B immunostaining in normal tissues.** A) Strong positive cytoplasmatic CELA3B immunostaining in acinar parenchyma of the pancreas. B). Strong positive cytoplasmatic CELA3B immunostaining in acinar parenchyma, ductal epithelium of individual pancreatic ducts (*) and focal weak cytoplasmatic staining in islets of Langerhans (**). C) Focal apical "membranous" staining within the brush border of the ileum as well as positive staining in extracellular mucin and debris. D) Focal apical "membranous" staining of the colon as well as strong positive staining in extracellular debris/feces.

undesired cross-reactivities of tested antibodies. A specific antibody reactivity in our experimental set-up is supported by the detection of a strong CELA3B immunostaining in the pancreas, the only organ with documented CELA3B RNA expression.

**Table 1. CELA3B immunostaining in tumors.**

| | Tumor entity | on TMA (n) | CELA3B #3682 immunostaining | | | |
| | | | analyzable (n) | negative (%) | weak (%) | moderate (%) | strong (%) |
|---|---|---|---|---|---|---|---|
| **Tumors of the skin** | Pilomatrixoma | 35 | 32 | 100.0 | 0.0 | 0.0 | 0.0 |
| | Basal cell carcinoma | 88 | 63 | 100.0 | 0.0 | 0.0 | 0.0 |
| | Benign nevus | 29 | 29 | 100.0 | 0.0 | 0.0 | 0.0 |
| | Squamous cell carcinoma of the skin | 90 | 90 | 100.0 | 0.0 | 0.0 | 0.0 |
| | Malignant melanoma | 48 | 45 | 100.0 | 0.0 | 0.0 | 0.0 |
| | Merkel cell carcinoma | 46 | 41 | 100.0 | 0.0 | 0.0 | 0.0 |
| **Tumors of the head and neck** | Squamous cell carcinoma of the larynx | 110 | 105 | 100.0 | 0.0 | 0.0 | 0.0 |
| | Squamous cell carcinoma of the pharynx | 60 | 59 | 100.0 | 0.0 | 0.0 | 0.0 |
| | Oral squamous cell carcinoma (floor of the mouth) | 130 | 129 | 100.0 | 0.0 | 0.0 | 0.0 |
| | Pleomorphic adenoma of the parotid gland | 50 | 39 | 100.0 | 0.0 | 0.0 | 0.0 |
| | Warthin tumor of the parotid gland | 104 | 99 | 100.0 | 0.0 | 0.0 | 0.0 |
| | Adenocarcinoma, NOS (Papillary Cystadenocarcinoma) | 14 | 12 | 100.0 | 0.0 | 0.0 | 0.0 |
| | Salivary duct carcinoma | 15 | 10 | 100.0 | 0.0 | 0.0 | 0.0 |
| | Acinic cell carcinoma of the salivary gland | 181 | 127 | 99.2 | 0.0 | 0.8 | 0.0 |
| | Adenocarcinoma NOS of the salivary gland | 109 | 65 | 100.0 | 0.0 | 0.0 | 0.0 |
| | Adenoid cystic carcinoma of the salivary gland | 180 | 91 | 98.9 | 0.0 | 1.1 | 0.0 |
| | Basal cell adenocarcinoma of the salivary gland | 25 | 22 | 100.0 | 0.0 | 0.0 | 0.0 |
| | Basal cell adenoma of the salivary gland | 101 | 86 | 100.0 | 0.0 | 0.0 | 0.0 |
| | Epithelial-myoepithelial carcinoma of the salivary gland | 53 | 50 | 100.0 | 0.0 | 0.0 | 0.0 |
| | Mucoepidermoid carcinoma of the salivary gland | 343 | 246 | 98.8 | 0.8 | 0.4 | 0.0 |
| | Myoepithelial carcinoma of the salivary gland | 21 | 18 | 100.0 | 0.0 | 0.0 | 0.0 |
| | Myoepithelioma of the salivary gland | 11 | 9 | 100.0 | 0.0 | 0.0 | 0.0 |
| | Oncocytic carcinoma of the salivary gland | 12 | 10 | 100.0 | 0.0 | 0.0 | 0.0 |
| | Polymorphous adenocarcinoma, low grade, of the salivary gland | 41 | 31 | 100.0 | 0.0 | 0.0 | 0.0 |
| | Pleomorphic adenoma of the salivary gland | 53 | 41 | 100.0 | 0.0 | 0.0 | 0.0 |
| **Tumors of the lung, pleura and thymus** | Adenocarcinoma of the lung | 196 | 176 | 100.0 | 0.0 | 0.0 | 0.0 |
| | Squamous cell carcinoma of the lung | 80 | 69 | 100.0 | 0.0 | 0.0 | 0.0 |
| | Small cell carcinoma of the lung | 16 | 16 | 100.0 | 0.0 | 0.0 | 0.0 |
| | Mesothelioma, epitheloid | 39 | 28 | 100.0 | 0.0 | 0.0 | 0.0 |
| | Mesothelioma, other types | 76 | 61 | 100.0 | 0.0 | 0.0 | 0.0 |
| | Thymoma | 29 | 29 | 100.0 | 0.0 | 0.0 | 0.0 |
| **Tumors of the female genital tract** | Squamous cell carcinoma of the vagina | 78 | 73 | 100.0 | 0.0 | 0.0 | 0.0 |
| | Squamous cell carcinoma of the vulva | 130 | 124 | 100.0 | 0.0 | 0.0 | 0.0 |
| | Squamous cell carcinoma of the cervix | 129 | 124 | 100.0 | 0.0 | 0.0 | 0.0 |
| | Endometrioid endometrial carcinoma | 236 | 225 | 100.0 | 0.0 | 0.0 | 0.0 |
| | Endometrial serous carcinoma | 82 | 73 | 100.0 | 0.0 | 0.0 | 0.0 |
| | Carcinosarcoma of the uterus | 48 | 46 | 100.0 | 0.0 | 0.0 | 0.0 |
| | Endometrial carcinoma, high grade, G3 | 13 | 13 | 100.0 | 0.0 | 0.0 | 0.0 |
| | Endometrial clear cell carcinoma | 8 | 8 | 100.0 | 0.0 | 0.0 | 0.0 |
| | Endometrioid carcinoma of the ovary | 110 | 93 | 100.0 | 0.0 | 0.0 | 0.0 |
| | Serous carcinoma of the ovary | 559 | 479 | 100.0 | 0.0 | 0.0 | 0.0 |
| | Mucinous carcinoma of the ovary | 96 | 71 | 100.0 | 0.0 | 0.0 | 0.0 |

*(Continued)*

**Table 1.** (Continued)

| | Tumor entity | on TMA (n) | CELA3B #3682 immunostaining | | | |
| | | | analyzable (n) | negative (%) | weak (%) | moderate (%) | strong (%) |
|---|---|---|---|---|---|---|---|
| | Clear cell carcinoma of the ovary | 50 | 26 | 100.0 | 0.0 | 0.0 | 0.0 |
| | Carcinosarcoma of the ovary | 47 | 42 | 100.0 | 0.0 | 0.0 | 0.0 |
| | Brenner tumor | 9 | 9 | 100.0 | 0.0 | 0.0 | 0.0 |
| **Tumors of the breast** | Invasive breast carcinoma of no special type | 80 | 71 | 100.0 | 0.0 | 0.0 | 0.0 |
| | Lobular carcinoma of the breast | 122 | 104 | 100.0 | 0.0 | 0.0 | 0.0 |
| | Medullary carcinoma of the breast | 15 | 13 | 100.0 | 0.0 | 0.0 | 0.0 |
| | Tubular carcinoma of the breast | 18 | 13 | 100.0 | 0.0 | 0.0 | 0.0 |
| | Mucinous carcinoma of the breast | 22 | 19 | 100.0 | 0.0 | 0.0 | 0.0 |
| | Phyllodes tumor of the breast | 50 | 44 | 100.0 | 0.0 | 0.0 | 0.0 |
| **Tumors of the digestive system** | Adenomatous polyp, low-grade dysplasia | 50 | 50 | 100.0 | 0.0 | 0.0 | 0.0 |
| | Adenomatous polyp, high-grade dysplasia | 50 | 49 | 100.0 | 0.0 | 0.0 | 0.0 |
| | Adenocarcinoma of the colon | 1882 | 1765 | 100.0 | 0.0 | 0.0 | 0.0 |
| | Gastric adenocarcinoma, diffuse type | 176 | 159 | 100.0 | 0.0 | 0.0 | 0.0 |
| | Gastric adenocarcinoma, intestinal type | 174 | 165 | 100.0 | 0.0 | 0.0 | 0.0 |
| | Gastric adenocarcinoma, mixed type | 62 | 55 | 100.0 | 0.0 | 0.0 | 0.0 |
| | Adenocarcinoma of the esophagus | 83 | 82 | 100.0 | 0.0 | 0.0 | 0.0 |
| | Squamous cell carcinoma of the esophagus | 75 | 71 | 100.0 | 0.0 | 0.0 | 0.0 |
| | Squamous cell carcinoma of the anal canal | 89 | 82 | 100.0 | 0.0 | 0.0 | 0.0 |
| | Cholangiocarcinoma | 113 | 108 | 100.0 | 0.0 | 0.0 | 0.0 |
| | Hepatocellular carcinoma | 50 | 50 | 100.0 | 0.0 | 0.0 | 0.0 |
| | Ductal adenocarcinoma of the pancreas | 612 | 448 | 100.0 | 0.0 | 0.0 | 0.0 |
| | Pancreatic/Ampullary adenocarcinoma | 89 | 69 | 100.0 | 0.0 | 0.0 | 0.0 |
| | Acinar cell carcinoma of the pancreas | 16 | 16 | 25.0 | 37.5 | 0.0 | 37.5 |
| | Gastrointestinal stromal tumor (GIST) | 50 | 48 | 100.0 | 0.0 | 0.0 | 0.0 |
| **Tumors of the urinary system** | Non-invasive papillary urothelial carcinoma, pTa G2 low grade | 177 | 141 | 100.0 | 0.0 | 0.0 | 0.0 |
| | Non-invasive papillary urothelial carcinoma, pTa G2 high grade | 141 | 117 | 100.0 | 0.0 | 0.0 | 0.0 |
| | Non-invasive papillary urothelial carcinoma, pTa G3 | 187 | 113 | 100.0 | 0.0 | 0.0 | 0.0 |
| | Urothelial carcinoma, pT2-4 G3 | 1206 | 818 | 100.0 | 0.0 | 0.0 | 0.0 |
| | Small cell neuroendocrine carcinoma of the bladder | 20 | 20 | 100.0 | 0.0 | 0.0 | 0.0 |
| | Clear cell renal cell carcinoma | 857 | 823 | 100.0 | 0.0 | 0.0 | 0.0 |
| | Papillary renal cell carcinoma | 255 | 232 | 100.0 | 0.0 | 0.0 | 0.0 |
| | Clear cell (tubulo) papillary renal cell carcinoma | 21 | 20 | 100.0 | 0.0 | 0.0 | 0.0 |
| | Chromophobe renal cell carcinoma | 131 | 122 | 100.0 | 0.0 | 0.0 | 0.0 |
| | Oncocytoma | 177 | 162 | 100.0 | 0.0 | 0.0 | 0.0 |
| **Tumors of the male genital organs** | Adenocarcinoma of the prostate, Gleason 3+3 | 83 | 83 | 100.0 | 0.0 | 0.0 | 0.0 |
| | Adenocarcinoma of the prostate, Gleason 4+4 | 80 | 80 | 100.0 | 0.0 | 0.0 | 0.0 |
| | Adenocarcinoma of the prostate, Gleason 5+5 | 85 | 85 | 100.0 | 0.0 | 0.0 | 0.0 |
| | Adenocarcinoma of the prostate (recurrence) | 258 | 251 | 100.0 | 0.0 | 0.0 | 0.0 |
| | Small cell neuroendocrine carcinoma of the prostate | 19 | 17 | 100.0 | 0.0 | 0.0 | 0.0 |
| | Seminoma | 621 | 603 | 100.0 | 0.0 | 0.0 | 0.0 |
| | Embryonal carcinoma of the testis | 50 | 44 | 100.0 | 0.0 | 0.0 | 0.0 |

(*Continued*)

**Table 1.** (Continued)

| | Tumor entity | on TMA (n) | CELA3B #3682 immunostaining | | | |
|---|---|---|---|---|---|---|
| | | | analyzable (n) | negative (%) | weak (%) | moderate (%) | strong (%) |
| | Yolk sak tumor | 50 | 43 | 100.0 | 0.0 | 0.0 | 0.0 |
| | Teratoma | 50 | 38 | 100.0 | 0.0 | 0.0 | 0.0 |
| | Squamous cell carcinoma of the penis | 80 | 79 | 100.0 | 0.0 | 0.0 | 0.0 |
| Tumors of endocrine organs | Adenoma of the thyroid gland | 114 | 111 | 100.0 | 0.0 | 0.0 | 0.0 |
| | Papillary thyroid carcinoma | 392 | 366 | 100.0 | 0.0 | 0.0 | 0.0 |
| | Follicular thyroid carcinoma | 154 | 154 | 100.0 | 0.0 | 0.0 | 0.0 |
| | Medullary thyroid carcinoma | 111 | 107 | 100.0 | 0.0 | 0.0 | 0.0 |
| | Anaplastic thyroid carcinoma | 45 | 43 | 100.0 | 0.0 | 0.0 | 0.0 |
| | Adrenal cortical adenoma | 50 | 44 | 100.0 | 0.0 | 0.0 | 0.0 |
| | Adrenal cortical carcinoma | 26 | 25 | 100.0 | 0.0 | 0.0 | 0.0 |
| | Phaeochromocytoma | 50 | 48 | 100.0 | 0.0 | 0.0 | 0.0 |
| | Appendix, neuroendocrine tumor (NET) | 22 | 15 | 100.0 | 0.0 | 0.0 | 0.0 |
| | Colorectal, neuroendocrine tumor (NET) | 12 | 8 | 100.0 | 0.0 | 0.0 | 0.0 |
| | Ileum, neuroendocrine tumor (NET) | 49 | 43 | 100.0 | 0.0 | 0.0 | 0.0 |
| | Lung, neuroendocrine tumor (NET) | 19 | 18 | 100.0 | 0.0 | 0.0 | 0.0 |
| | Pancreas, neuroendocrine tumor (NET) | 97 | 81 | 100.0 | 0.0 | 0.0 | 0.0 |
| | Colorectal, neuroendocrine carcinoma (NEC) | 12 | 7 | 100.0 | 0.0 | 0.0 | 0.0 |
| | Gallbladder, neuroendocrine carcinoma (NEC) | 4 | 3 | 100.0 | 0.0 | 0.0 | 0.0 |
| | Pancreas, neuroendocrine carcinoma (NEC) | 14 | 14 | 100.0 | 0.0 | 0.0 | 0.0 |
| Tumors of haemotopoetic and lymphoid tissues | Hodgkin Lymphoma | 103 | 100 | 100.0 | 0.0 | 0.0 | 0.0 |
| | Small lymphocytic lymphoma, B-cell type (B-SLL/B-CLL) | 50 | 50 | 100.0 | 0.0 | 0.0 | 0.0 |
| | Diffuse large B cell lymphoma (DLBCL) | 114 | 114 | 100.0 | 0.0 | 0.0 | 0.0 |
| | Follicular lymphoma | 88 | 88 | 100.0 | 0.0 | 0.0 | 0.0 |
| | T-cell Non Hodgkin lymphoma | 24 | 24 | 100.0 | 0.0 | 0.0 | 0.0 |
| | Mantle cell lymphoma | 18 | 18 | 100.0 | 0.0 | 0.0 | 0.0 |
| | Marginal zone lymphoma | 16 | 16 | 100.0 | 0.0 | 0.0 | 0.0 |
| | Diffuse large B-cell lymphoma (DLBCL) in the testis | 16 | 16 | 100.0 | 0.0 | 0.0 | 0.0 |
| | Burkitt lymphoma | 5 | 3 | 100.0 | 0.0 | 0.0 | 0.0 |
| Tumors of soft tissue and bone | Tenosynovial giant cell tumor | 45 | 43 | 100.0 | 0.0 | 0.0 | 0.0 |
| | Granular cell tumor | 53 | 43 | 100.0 | 0.0 | 0.0 | 0.0 |
| | Leiomyoma | 50 | 47 | 100.0 | 0.0 | 0.0 | 0.0 |
| | Leiomyosarcoma | 87 | 87 | 100.0 | 0.0 | 0.0 | 0.0 |
| | Liposarcoma | 132 | 121 | 100.0 | 0.0 | 0.0 | 0.0 |
| | Malignant peripheral nerve sheath tumor (MPNST) | 13 | 11 | 100.0 | 0.0 | 0.0 | 0.0 |
| | Myofibrosarcoma | 26 | 26 | 100.0 | 0.0 | 0.0 | 0.0 |
| | Angiosarcoma | 73 | 67 | 100.0 | 0.0 | 0.0 | 0.0 |
| | Angiomyolipoma | 91 | 88 | 100.0 | 0.0 | 0.0 | 0.0 |
| | Dermatofibrosarcoma protuberans | 21 | 18 | 100.0 | 0.0 | 0.0 | 0.0 |
| | Ganglioneuroma | 14 | 14 | 100.0 | 0.0 | 0.0 | 0.0 |
| | Kaposi sarcoma | 8 | 5 | 100.0 | 0.0 | 0.0 | 0.0 |
| | Neurofibroma | 117 | 116 | 100.0 | 0.0 | 0.0 | 0.0 |
| | Sarcoma, not otherwise specified (NOS) | 74 | 70 | 100.0 | 0.0 | 0.0 | 0.0 |

(*Continued*)

**Table 1.** (Continued)

| | Tumor entity | on TMA (n) | CELA3B #3682 immunostaining | | | | |
|---|---|---|---|---|---|---|---|
| | | | analyzable (n) | negative (%) | weak (%) | moderate (%) | strong (%) |
| | Paraganglioma | 41 | 41 | 100.0 | 0.0 | 0.0 | 0.0 |
| | Ewing sarcoma | 23 | 16 | 100.0 | 0.0 | 0.0 | 0.0 |
| | Rhabdomyosarcoma | 6 | 6 | 100.0 | 0.0 | 0.0 | 0.0 |
| | Schwannoma | 121 | 118 | 100.0 | 0.0 | 0.0 | 0.0 |
| | Synovial sarcoma | 12 | 11 | 100.0 | 0.0 | 0.0 | 0.0 |
| | Osteosarcoma | 43 | 36 | 100.0 | 0.0 | 0.0 | 0.0 |
| | Chondrosarcoma | 38 | 17 | 100.0 | 0.0 | 0.0 | 0.0 |

(Note: All positive cases are highlighted in grey).

Given the lack of documented CELA3B RNA expression in other organs the additional (focal) CELA3B immunostaining of surface membranes of the small intestine and the colorectum was not expected. True intestinal CELA3B staining is, however, supported by similar colon surface staining shown for antibody HPA045650 used in the human protein atlas [20]. Although the known function of CELA3B is not coherent with a role as a membrane protein, true membranous CELA3B expression in intestinal surface epithelium cannot be excluded. These surface epithelial cells may constitute sufficiently small subsets of the total amount of cells in the colon to remain undetected in RNA analyses. As CELA3B is secreted into the intestine in large quantities, it is also possible, that these intestinal membranous surface staining is due to secreted pancreatic CELA3B which adheres to the intestinal epithelium. This is all the more conceivable as CELA3B suffers no proteolytic degradation during intestinal transit and can be detected in high concentrations in the stool [21]. Hence the detection of CELA3B is clinically used to assess exocrine pancreatic insufficiency [22]. ScheBo Pancreatic Elastase 1 Stool Test (ScheBo Biotech, Giessen, Germany) is a widely used assay which detects CELA3B in the stool with an ELISA test.

The successful analysis of 13,223 cancers from 132 different tumor entities revealed that CELA3B expression was strikingly linked to acinar cell carcinoma of the pancreas (12 of 16 cases positive). Acinar cell carcinoma is a rare subtype of pancreatic cancer derived from acinar cells and making up for about 1–2% of all pancreatic neoplasms [23, 24]. Given the unequivocal CELA3B staining in normal epithelial cells from excretory ducts and the previous observation of CELA3B upregulation in pancreatic intraepithelial neoplasia and pancreatic carcinomas of rats [25] this finding was not expected. The complete lack of CELA3B expressing neoplastic cells in 448 ductal carcinomas of the pancreas strongly argues against a relevant CELA3B protein production in these cells, however. Since large quantities of CELA3B protein are transported through the pancreatic ducts, it must be considered that CELA3B can be absorbed by normal excretory duct cells to some extent. Alternatively, CELA3B staining of non-acinar cells in the pancreas may reflect a contamination artifact caused by diffusion of the highly abundant CELA3B protein from acinar cells to adjacent structures. Since all our pancreatic samples were from pancreatectomies, CELA3B diffusion might be facilitated by—even minimal—acinar cell damage due to hypoxia occurring during the surgical removal of the pancreas.

CELA3B closely interacts with Carboxypeptidase A1 (CPA1) under physiological conditions. ProCELA3B forms complexes with proCPA1, which increases binding activity of the inhibitory activation peptide of procarboxpeptidases and thereby stabilizes the zymogen state

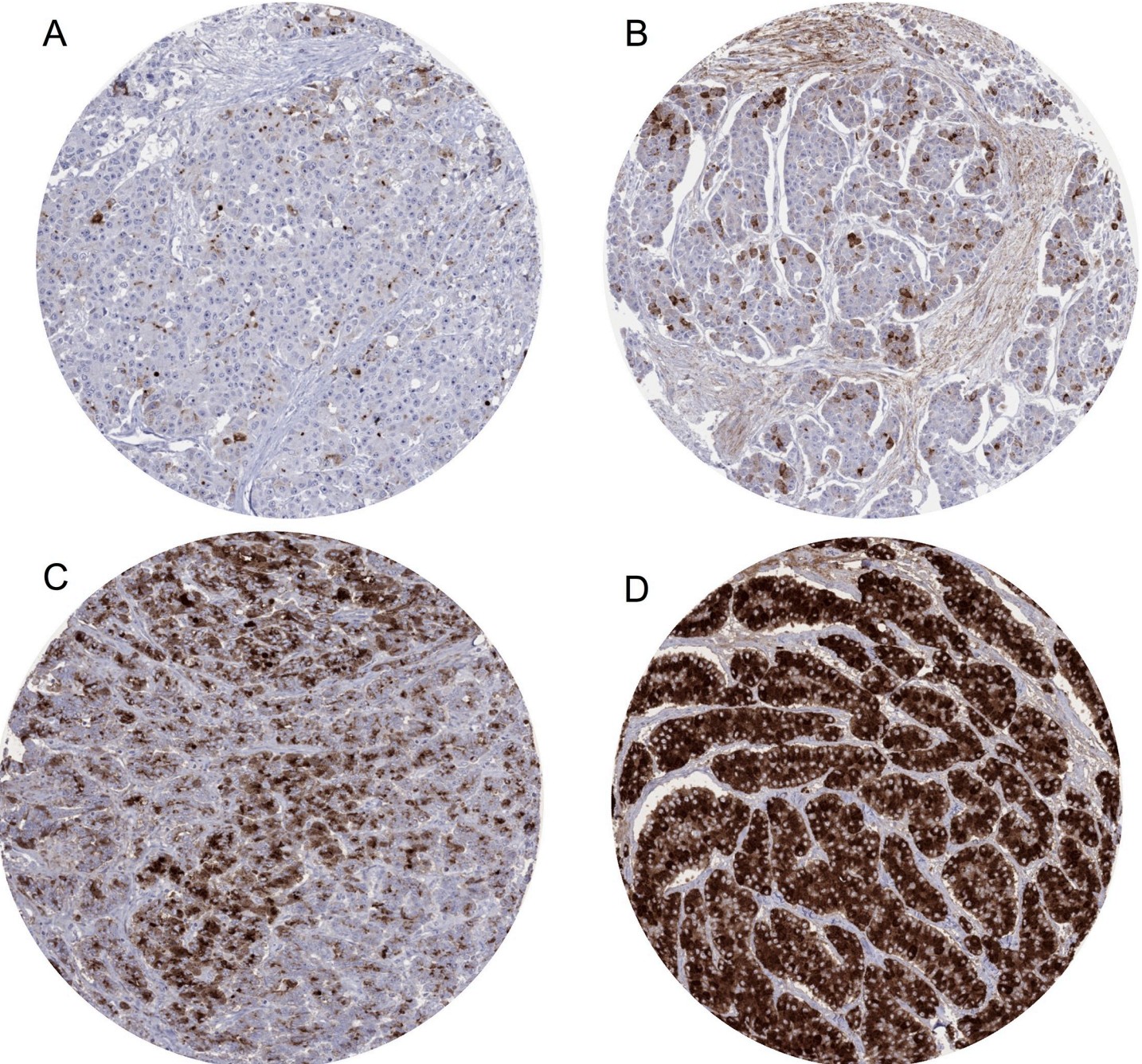

**Fig 2. Various patterns of CELA3B immunostaining in acinar cell carcinoma of the pancreas.** A) and B) Focal weak to strong cytoplasmatic CELA3B in individual tumor cells. C) Patchy moderate to strong cytoplasmatic immunostaining. D) Homogenous strong cytoplasmatic staining.

[5]. This interaction fits well with the significant association of CELA3B and CPA1 expression in acinar cell carcinomas. That CELA3B immunostaining was only weak in a fraction of our acinar cell carcinomas is consistent with earlier data describing a reduced CELA3B expression in pancreatic cancer cell lines and tissues as compared to adjacent pancreatic normal tissues [26]. Reduced CELA3B (and CPA1) expression in acinar cell carcinomas may reflect cellular de-differentiation and could therefore potentially be linked to unfavorable patient prognosis.

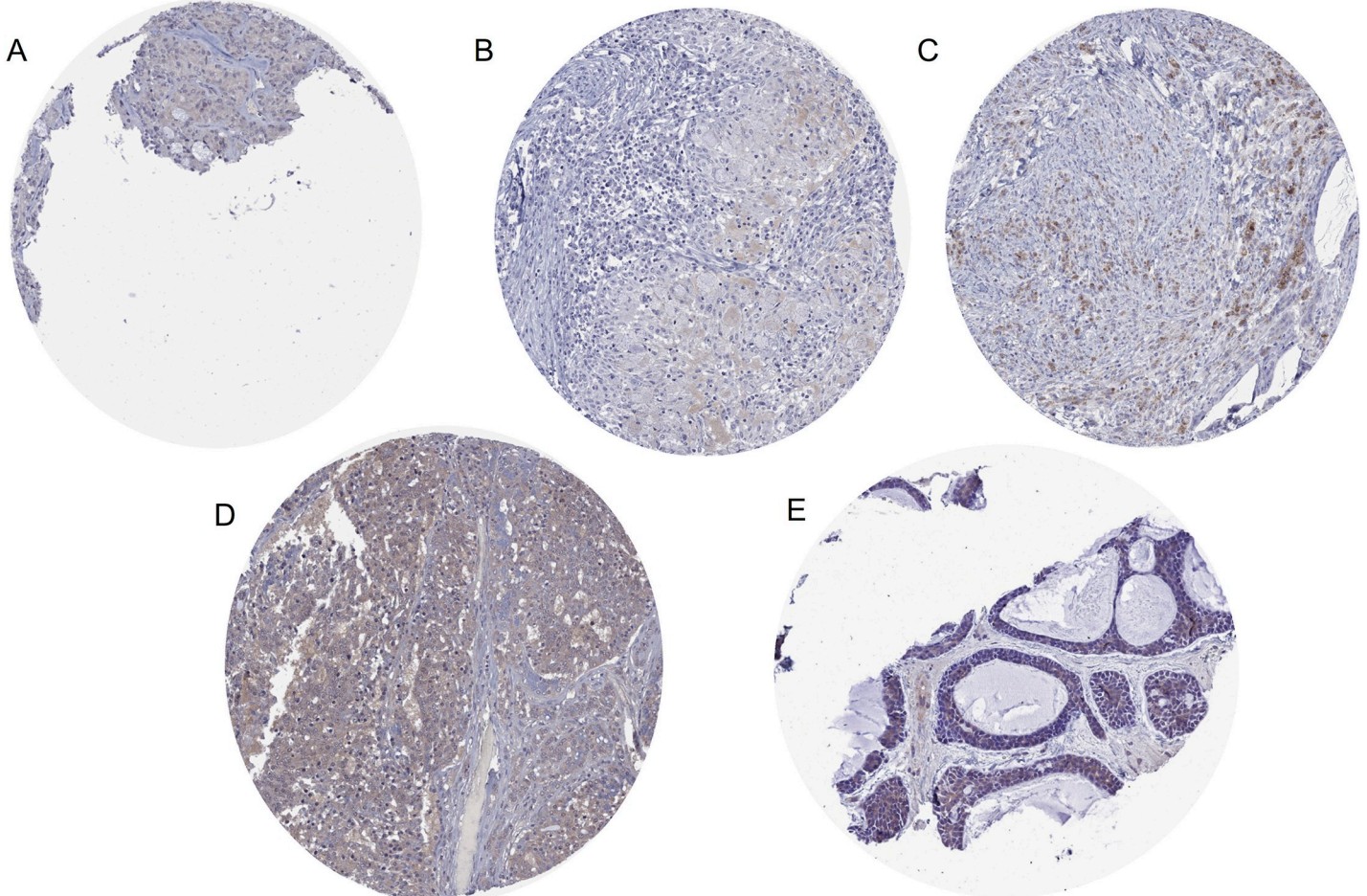

**Fig 3. Equivocal CELA3B immunostaining in non-pancreatic tumors.** A)-C) Focal weak to moderate cytoplasmatic fine granular staining in mucoepidermoid carcinoma of the salivary gland. D) Homogenous weak "brown staining" in acinic cell carcinoma of the salivary gland. E) Adenoid Cystic Carcinoma.

Unequivocal CELA3B positivity was not seen in tumors other than pancreatic acinar cell carcinomas in this study. The 5 CELA3B positive non-pancreatic cancers included several salivary gland tumors with weak to moderate cytoplasmatic granular staining. It is of note that salivary glands and the pancreatic gland share histological and functional features such as the organization into ductal and acinar cells and the secretion of digestive enzymes [27]. Moreover, both organs appear to interact under certain conditions. For example, salivary gland function is often impaired in patients suffering from diabetes mellitus [28], and deficiencies in epidermal growth factor secretion of the salivary gland has been linked to symptoms of diabetes [29]. It is quite possible that the pathological activation of CELA3B in some salivary gland tumors is another expression of similarities between the two organs.

The high specificity of CELA3B immunostaining for pancreatic acinar carcinoma is also consistent with RNA expression data summarized in the TCGA/ICGC database (https://www. cancer.gov/tcga). Among 17 different cancer categories (n = 7,932 samples), significant CELA3B expression was only found in 144 of 176 (82%) analyzed pancreatic cancers. Based on our data, it can be assumed that these CELA3B positive cases include acinar cell carcinomas and other pancreatic neoplasms of which the analyzed samples also contained normal pancreatic tissue.

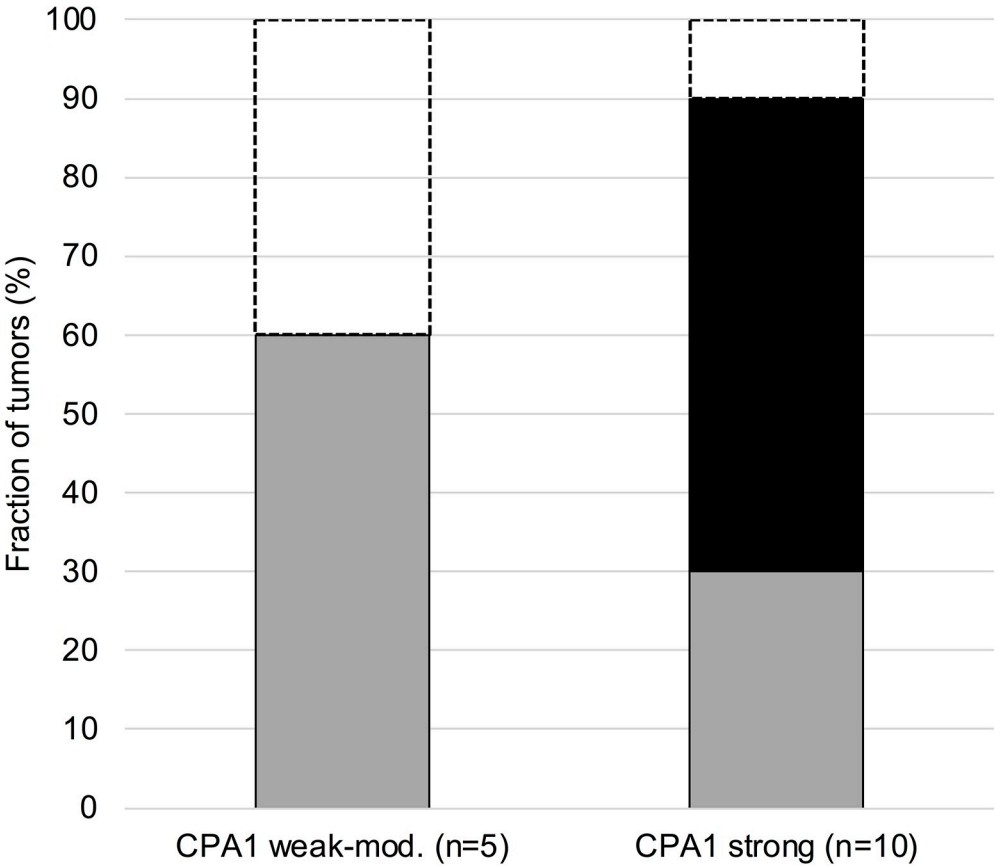

**Fig 4. Comparison of CELA3B immunostaining with CPA1 immunostaining in acinar cell carcinoma of the pancreas.**

Given its high specificity, CELA3B immunohiostochemistry may represent a useful tool for supporting the difficult diagnosis of acinar cell carcinoma, although the positivity rate was only 75%. Molecular markers are essential for the difficult distinction of these tumors from neuroendocrine neoplasms or ductal adenocarcinomas of the pancreas because acinar cell carcinomas can show various architectural patterns apart from the "classic" acinar pattern. Even "experts" in gastrointestinal pathology will only rarely come across these tumors. In a retrospective study at John Hopkins only 14 acinar cell carcinomas were identified over a period of 18 years [30]. Because of the rarity of this tumor, unusual patterns are regularly misclassified by pathologists [31–33]. For example, Basturk et al reclassified 17 of 107 tumors initially diagnosed as poorly differentiated neuroendocrine carcinomas as pure acinar cell carcinoma or mixed acinar-neuroendocrine carcinoma [31]. Studies are needed to investigate whether the additional use of CELA3B will increase the diagnostic precision that can be achieved by using CPA1 [19], chymotrypsin [34], trypsin [34], or bcl10 [35] for diagnosing acinar cell carcinoma of the pancreas.

In conclusion, our data show a high specificity of CELA3B immunostaining for acinar cell differentiation in the pancreas. As a part of an antibody panel, CELA3B

immunohistochemistry may represent a useful diagnostic marker for confirming the difficult diagnosis of pancreatic acinar cell carcinoma.

## Acknowledgments

We are grateful to Melanie Witt, Inge Brandt, Maren Eisenberg, and Sünje Seekamp for excellent technical assistance.

## Author Contributions

**Conceptualization:** Ria Uhlig, Guido Sauter, Stefan Steurer, Rainer Krech, Ronald Simon, Till Krech, Frank Jacobsen.

**Data curation:** Ria Uhlig, Nina Bröker, Sören Weidemann, Natalia Gorbokon, Anne Menz, Franziska Büscheck, Andreas M. Luebke, Devita Putri, Martina Kluth, Andrea Hinsch, Maximilian Lennartz, Viktor Reiswich, Doris Höflmayer, Christoph Fraune, Katharina Möller, Christian Bernreuther, Patrick Lebok, Guido Sauter, Sarah Minner, Stefan Steurer, Eike Burandt, Rainer Krech, David Dum, Andreas Marx, Ronald Simon, Till Krech, Till S. Clauditz, Frank Jacobsen.

**Formal analysis:** Nina Bröker, Sören Weidemann, Natalia Gorbokon, Anne Menz, Franziska Büscheck, Andreas M. Luebke, Devita Putri, Martina Kluth, Claudia Hube-Magg, Andrea Hinsch, Maximilian Lennartz, Viktor Reiswich, Doris Höflmayer, Christoph Fraune, Katharina Möller, Christian Bernreuther, Patrick Lebok, Guido Sauter, Sarah Minner, Eike Burandt, Rainer Krech, David Dum, Andreas Marx, Ronald Simon, Till Krech, Till S. Clauditz, Frank Jacobsen.

**Methodology:** Ria Uhlig, Natalia Gorbokon, Maximilian Lennartz, Viktor Reiswich, Christoph Fraune, Sarah Minner, Till S. Clauditz, Frank Jacobsen.

**Supervision:** Ria Uhlig, Guido Sauter, Ronald Simon, Till S. Clauditz, Frank Jacobsen.

**Validation:** Claudia Hube-Magg.

**Writing – original draft:** Ria Uhlig, Guido Sauter, Ronald Simon, Till S. Clauditz, Frank Jacobsen.

**Writing – review & editing:** Ria Uhlig, Guido Sauter, Ronald Simon, Till S. Clauditz, Frank Jacobsen.

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
