## [Decision Letter · Decision Letter 0]

10 Oct 2022

PONE-D-22-21974CELA3B immunostaining is a highly specific marker for acinar cell carcinoma of the pancreasPLOS ONE

Dear Dr. Simon,

Thank you for submitting your manuscript to PLOS ONE. After careful consideration, we feel that it has merit but does not fully meet PLOS ONE’s publication criteria as it currently stands. Therefore, we invite you to submit a revised version of the manuscript that addresses the points raised during the review process.

We look forward to receiving your revised manuscript.

Kind regards,

Shuai Ren

Academic Editor

PLOS ONE

Journal Requirements:

2. Our staff editors have determined that your manuscript is likely within the scope of our Early Detection, Screening and Diagnosis of Cancer Call for Papers. This editorial initiative is headed by in-house PLOS editors. This Call for Papers aims to explore recent advances in the early detection of cancer and implications of these advances for patient survival. Additional information can be found on our announcement page: https://collections.plos.org/call-for-papers/early-detection-screening-and-diagnosis-of-cancer/

If you would like your manuscript to be considered for this collection, please let us know in your cover letter and we will ensure that your paper is treated as if you were responding to this call.  Please note that being considered for the Call for Papers does not require additional peer review beyond the journal’s standard process and will not delay the publication of your manuscript if it is accepted by PLOS ONE. If you would prefer to remove your manuscript from collection consideration, please specify this in the cover letter.

   "The CELA3B antibody clone MSVA-410M was provided from MS Validated Antibodies GmbH (owned by a family member of GS)."

Additional Editor Comments :

Congratulations on the good work! Only minor revision is recommeded!

Reviewers' comments:

Reviewer's Responses to Questions

**Comments to the Author**

1. Is the manuscript technically sound, and do the data support the conclusions?

Reviewer #1: Yes

Reviewer #2: Partly

2. Has the statistical analysis been performed appropriately and rigorously? 

Reviewer #1: Yes

Reviewer #2: Yes

3. Have the authors made all data underlying the findings in their manuscript fully available?

Reviewer #1: Yes

Reviewer #2: Yes

4. Is the manuscript presented in an intelligible fashion and written in standard English?

Reviewer #1: Yes

Reviewer #2: Yes

5. Review Comments to the Author

Reviewer #1: This is an extensive IHC analysis of an impressive number of human tumors and normal tissues studying the expression of chymotrypsin-like elastase family member 3B (CELA3B), a pancreatic enzyme with digestive function in the intestine. Since previous RNA analyses had suggest that CELA3B expression is limited to the pancreas, the authors wanted to explore the diagnostic utility of CELA3B IHC for the distinction of pancreatic from extrapancreatic cancers and for the distinction of the rare acinar cell carcinomas from ductal adenocarcinomas.

Using TMAs, CELA3B expression was successfully analyzed by IHC in 13,223 tumor samples from 132 tumor types/subtypes and 76 different normal tissues. In principle, CELA3B staining was limited to acinar cells and to a fraction of ductal cells of the pancreas. In tumors, CELA3B was only expressed in 12 of 16 (75%) pancreatic acinar cell carcinomas, including 6 cases with strong staining as well as in 5 of 13,207 other tumors (0.04%). The latter included a few cases of salivary gland tumors. The authors conclude that the CELA3B IHC data shows good sensitivity (75%) and high specificity (99.9%) for diagnosing acinar cell carcinoma of the pancreas.

This is a solid and well-written paper exploring the possible use of CELA3B IHC for the diagnosis of a rare and difficult to diagnose subset of pancreatic cancers. The results are clear and convincing and the study is the first to use IHC to study the expression of CELA3B in tumors. The authors have long and well-documented experience in producing and analyzing TMAs. I only have a few minor comments.

To resolve the issue of CELA3B staining in non-acinar pancreatic cells, I suggest the authors use RNA in situ hybridization to look for mRNA expression in these cells.

The authors should discuss the observation of CELA3B staining in the few salivary gland tumors.

The quality of the immunostainings in Fig 3 (“non-pancreatic” tumors with positive CELA3B staining) could be improved.

Reviewer #2: Immunohistochemistry in a tissue microarray format (TMA) was used to analyze the CELA3B expression in 13,223 tumor samples from 132 different tumor types and subtypes as well as 8 samples each of 76 different normal tissue types. The authors have found that the CELA3B immunostaining was seen in 12 of 16 (75%) acinar cell carcinoma of the pancreas including 6 cases with strong staining (37.5%) as well as in 5 of 13,207 other tumors (0.04%). This study is of important significance for the clinical pathological diagnosis. However, evidence for good sensitivity (75%) of CELA3B immunohistochemistry for diagnosing acinar cell carcinoma of the pancreas is not sufficient. Here are some questions before this article was published:

1. Most importantly, the number of positive cases of acinar cell carcinoma of the pancreas, which was only 16, was low not only in the overall database (13,223 cases) but also in comparison. More cases of acinar cell carcinoma of the pancreas are needed in order to draw a conclusion.

2. How do the authors draw conclusions of high sensitivity (99.9%) of CELA3B immunohistochemistry for diagnosing acinar cell carcinoma of the pancreas?

3. Although CELA3B immunostaining was seen in 12 of 16 (75%) acinar cell carcinoma of the pancreas，but was essentially negative in the remaining 4 cases (20%)，suggesting that the absence of CELA3B immunostaining is a good predictor of acinar cell carcinoma of the pancreas.

6. PLOS authors have the option to publish the peer review history of their article (what does this mean?). If published, this will include your full peer review and any attached files.

Reviewer #1: No

Reviewer #2: No

---

## [Author Response · Author response to Decision Letter 0]

9 Feb 2023

Dear Reviewers and dear Prof Chenette, 

Please find enclosed our revised manuscript entitled “CELA3B complements CPA1 in the diagnosis of acinar cell carcinoma of the pancreas”. We would like to thank the reviewers for their comments on how to improve the manuscript. Please find below our point-by-point responses to the academic editor and the reviewers.

In reply to the Academic Editor

Reply: We have now the update the PLOS ONE's style requirements in our manuscript.

2. If you would like your manuscript to be considered for this collection (Early Detection, Screening and Diagnosis of Cancer), please let us know in your cover letter and we will ensure that your paper is treated as if you were responding to this call. 

Reply: Please consider our manuscript for this collection.

 "The CELA3B antibody clone MSVA-410M was provided from MS Validated Antibodies GmbH (owned by a family member of GS)."

Please confirm that this does not alter your adherence to all PLOS ONE policies on sharing data and materials, 

Reply: We have added the statement to the letter (see below) and to the manuscript. 

“The CELA3B antibody clone MSVA-410M was provided from MS Validated Antibodies GmbH (owned by a family member of GS). This does not alter our adherence to PLOS ONE policies on sharing data and materials.”

4. In your Data Availability statement, you have not specified where the minimal data set underlying the results described in your manuscript can be found. We will update your Data Availability statement to reflect the information you provide in your cover letter.

Reply: We have updated our Data Availability statement in the editorial manager:

“Raw data are available upon reasonable request. All data relevant to the study are included in the article.”

Reply: We have removed it from the other sections.

6. Please review your reference list to ensure that it is complete and correct. 

Reply: We have checked our reference list.

In reply to the reviewers

In reply to reviewer #1:

1) To resolve the issue of CELA3B staining in non-acinar pancreatic cells, I suggest the authors use RNA in situ hybridization to look for mRNA expression in these cells.

Reply: The reviewer recommended adding experimental proof for CELA3B expression in salivary gland cancers. In reply, we have now more prominently highlighted RNA data that support CELA3B RNA expression in various tumor types the introduction on page 3, lines 64-66. We did not perform RNA in-situ hybridization experiments because we made dismal experiences with these technique in tissue microarray spots from formalin-fixed tissues.

2) The authors should discuss the observation of CELA3B staining in the few salivary gland tumors.

Reply: We have discussed similarities between salivary glands and the pancreatic gland that might explain occasional CELA3B staining in salivary gland tumors on page 8, lines 187-195.

3) The quality of the immunostainings in Fig 3 (“non-pancreatic” tumors with positive CELA3B staining) could be improved.

Reply: The poor quality is only due to the file compression for the review. The original images are high resolution.

In reply to reviewer #2:

1) Most importantly, the number of positive cases of acinar cell carcinoma of the pancreas, which was only 16, was low not only in the overall database (13,223 cases) but also in comparison. More cases of acinar cell carcinoma of the pancreas are needed in order to draw a conclusion.

Reply: We agree with the reviewer that more cases would be desirable. However, acinus cell carcinoma is a very rare condition and more cases were not available.

2-3) How do the authors draw conclusions of high sensitivity (99.9%) of CELA3B immunohistochemistry for diagnosing acinar cell carcinoma of the pancreas? Although CELA3B immunostaining was seen in 12 of 16 (75%) acinar cell carcinoma of the pancreas, but was essentially negative in the remaining 4 cases (20%)，suggesting that the absence of CELA3B immunostaining is a good predictor of acinar cell carcinoma of the pancreas.

Reply: We now provide detailed data on tumors with and without CELA3B staining in the results section (page 5-6, lines 128-130) that allow for calculating sensitivity and specificity.

We are grateful for the time and effort taken to review our manuscript. 

Sincerely,

Ronald Simon

---

## [Decision Letter · Decision Letter 1]

7 Jun 2023

CELA3B immunostaining is a highly specific marker for acinar cell carcinoma of the pancreas

PONE-D-22-21974R1

Dear Dr. Simon,

We’re pleased to inform you that your manuscript has been judged scientifically suitable for publication and will be formally accepted for publication once it meets all outstanding technical requirements.

Kind regards,

Shuai Ren

Academic Editor

PLOS ONE

Additional Editor Comments (optional):

The final version is ready for publication now.

Reviewers' comments:

Reviewer's Responses to Questions

**Comments to the Author**

1. If the authors have adequately addressed your comments raised in a previous round of review and you feel that this manuscript is now acceptable for publication, you may indicate that here to bypass the “Comments to the Author” section, enter your conflict of interest statement in the “Confidential to Editor” section, and submit your "Accept" recommendation.

Reviewer #1: (No Response)

Reviewer #2: All comments have been addressed

2. Is the manuscript technically sound, and do the data support the conclusions?

Reviewer #1: Yes

Reviewer #2: Yes

3. Has the statistical analysis been performed appropriately and rigorously? 

Reviewer #1: Yes

Reviewer #2: Yes

4. Have the authors made all data underlying the findings in their manuscript fully available?

Reviewer #1: Yes

Reviewer #2: Yes

5. Is the manuscript presented in an intelligible fashion and written in standard English?

Reviewer #1: Yes

Reviewer #2: Yes

6. Review Comments to the Author

Reviewer #1: 2) The authors should discuss the observation of CELA3B staining in the few salivary gland tumors.

Authors reply: We have discussed similarities between salivary glands and the pancreatic gland that might explain occasional CELA3B staining in salivary gland tumors on page 8, lines 187-195.

As far as I can see there is no discussion or mentioning of salivary gland tumors on page 8, lines 187-195. This part only concerns colorectal cancers. Please add the discussion about CELA3B staining in salivary gland tumors.

Reviewer #2: (No Response)

7. PLOS authors have the option to publish the peer review history of their article (what does this mean?). If published, this will include your full peer review and any attached files.

Reviewer #1: No

Reviewer #2: No

---

## [Editor Report · Acceptance letter]

16 Jun 2023

PONE-D-22-21974R1 

CELA3B immunostaining is a highly specific marker for acinar cell carcinoma of the pancreas 

Dear Dr. Simon:

I'm pleased to inform you that your manuscript has been deemed suitable for publication in PLOS ONE. Congratulations! Your manuscript is now with our production department. 

Kind regards, 

on behalf of

Dr. Shuai Ren 

Academic Editor

PLOS ONE